# Gut Microbiota-Mediated Inflammation and Gut Permeability in Patients with Obesity and Colorectal Cancer

**DOI:** 10.3390/ijms21186782

**Published:** 2020-09-16

**Authors:** Lidia Sánchez-Alcoholado, Rafael Ordóñez, Ana Otero, Isaac Plaza-Andrade, Aurora Laborda-Illanes, José Antonio Medina, Bruno Ramos-Molina, Jaime Gómez-Millán, María Isabel Queipo-Ortuño

**Affiliations:** 1Unidad de Gestión Clínica Intercentros de Oncología Médica, Hospitales Universitarios Regional y Virgen de la Victoria, Instituto de Investigación Biomédica de Málaga (IBIMA)-CIMES-UMA, 29010 Málaga, Spain; l.s.alcoholado@gmail.com (L.S.-A.); isaacplazaandrade@gmail.com (I.P.-A.); auroralabordaillanes@gmail.com (A.L.-I.); 2Facultad de Medicina, Universidad de Málaga, Campus de Teatinos s/n, 29071 Málaga, Spain; 3Unidad de Gestión Clínica de Oncología Radioterápica, Hospital Universitario Virgen de la Victoria, Instituto de Investigación Biomédica de Málaga (IBIMA), 29010 Málaga, Spain; rafaelordm@gmail.com (R.O.); ana.otero.rom@gmail.com (A.O.); jmedinacarmona@gmail.com (J.A.M.); 4Grupo de Cirugía Digestiva, Endocrina y Transplante de Órganos Abdominales, Instituto Murciano de Investigación Biosanitaria (IMIB-Arrixaca), 30120 Murcia, Spain; brunoramosmolina@gmail.com

**Keywords:** colorectal cancer, gut microbiota, obesity, inflammation, gut permeability, TMAO

## Abstract

Obesity is considered an important factor that increases the risk of colorectal cancer (CRC). So far, the association of gut microbiota with both obesity and cancer has been described independently. Nevertheless, a specific obesity-related microbial profile linked to CRC development has not been identified. The aim of this study was to determine the gut microbiota composition in fecal samples from CRC patients with (OB-CRC) and without obesity (L-CRC) compared to the microbiota profile present in non-obese healthy controls (L-HC), in order to unravel the possible relationship between gut microbiota and microbial-derived metabolite trimethylamine N-oxide (TMAO), the inflammatory status, and the intestinal permeability in the context of obesity-associated CRC. The presence of obesity does not induce significant changes in the diversity and richness of intestinal bacteria of CRC patients. Nevertheless, OB-CRC patients display a specific gut microbiota profile characterized by a reduction in butyrate-producing bacteria and an overabundance of opportunistic pathogens, which in turn could be responsible, at least in part, for the higher levels of proinflammatory cytokine IL-1β, the deleterious bacterial metabolite TMAO, and gut permeability found in these patients. These results suggest a possible role of obesity-related gut microbiota in the development of CRC, which could give new clues for the design of new diagnostic tools for CRC prevention.

## 1. Introduction

Colorectal cancer (CRC) is the third most incident cancer type worldwide [1]. The increased incidence of CRC in the last decades could be a consequence of the modern lifestyle, which is associated with age, obesity, the intake of high-fat/low-fiber diets, and the lack of physical activity [2,3]. It has been reported that obesity, particularly central obesity, is one of the most significant predisposing factors for numerous cancers and chronic diseases [4]. Moreover, it has been shown that obesity is a meaningful contributor to CRC and it has been considered as a poor prognosis factor in cancer development. Several epidemiological studies have demonstrated that adult obesity increases the risk of colon cancer 1.2–2-fold, with obesity accounting for 14–35% of total colon cancer incidence [5,6]. Although many factors could contribute to obesity-driven tumorigenesis, few studies have addressed whether the relationship between obesity and CRC could be mediated by alterations in the gut microbiota.

Obesity-associated dysbiosis may result in physiological changes that could increase the risk of cancer [7]. In fact, several studies have showed that specific bacterial taxa linked to obesity could play a role in the etiology of CRC [8,9,10,11,12]. Furthermore, intestinal inflammation and genotoxin-induced DNA damage of intestinal cells have been proposed as the possible mechanisms responsible for the role of microbial dysbiosis in carcinogenesis [13]. Chronic low-grade inflammation is a hallmark of both obesity and CRC etiology. Indeed, a causal relationship between chronic inflammation and CRC carcinogenesis has been already well-established [14]. On the other hand, several studies have shown that gut microbiota is an essential factor in driving inflammation in the colon, and this inflammatory environment is related to CRC development [15]. Moreover, obesity-induced alterations in intestinal barrier permeability may have an additional influence on CRC development. This obesity-induced barrier disruption induces a metabolic endotoxemia that synergizes with pre-existing adipose tissue inflammation to further increase tumor-promoting systemic inflammation and contribute to the production of pro-inflammatory cytokines [16].

Trimethylamine N-oxide (TMAO), a gut microbiota-derived metabolite of dietary choline and L-carnitine, which is obtained from red meat and other animal foods, has been associated with an elevated risk of CRC, as well as to obesity, cardiovascular disease, and diabetes [17,18,19,20,21]. Bae et al. showed that plasma TMAO levels were positively associated with CRC risk in a nested case-control study among women in the United States [19]. Another study using genome-wide analysis revealed that TMAO was genetically associated with CRC. Therefore, TMAO could be an important intermediate marker linking gut microbiota metabolism and CRC pathogenesis, although the molecular pathway that links TMAO to CRC remains to be characterized [20]. Recent evidence has suggested that inflammation could be a possible factor that provides a link between TMAO and cancer, although other mechanisms such as oxidative stress, DNA damage, and disruption in protein folding might also play a role [22].

Based on these data, the aim of this study was to determine the intestinal microbiota composition in fecal samples from CRC patients with and without obesity compared to the microbiota present in non-obese healthy controls, in order to unravel the possible association between the gut microbiota and microbial-derived metabolite TMAO, the inflammatory status, and the intestinal permeability in the context of obesity-associated CRC.

## 2. Results

The clinical and anthropometric characteristics of study subjects are summarized in Table 1. No significant differences in age, gender, race, alcohol, and tobacco consumption, constipation, total cholesterol, triglycerides, high density lipoprotein cholesterol (HDL-cholesterol), LDL-cholesterol, fasting glucose, and HbA1c levels were found between study groups (*p* > 0.05). As expected, significant differences were found in BMI between the CRC patients with obesity (OB-CRC) and non-obese healthy controls (L-HC). In addition, significantly higher levels of serum proinflammatory IL-1β and TMAO, and lower levels of anti-inflammatory IL-10 were found in the OB-CRC and non-obese CRC (L-CRC) groups compared to L-HC individuals. Moreover, these differences were even more pronounced when the OB-CRC patients were compared to L-HC subjects. No significant differences with respect to tumor stage and grade of tumor differentiation (*p* > 0.05) were found between L-CRC and OB-CRC groups.

### 2.1. Richness and Diversity of Fecal Microbiota

A total of 5,206,881 good quality 16S rRNA gene sequences (average of 80,105.86 ± 34,581.61 sequences per sample) were obtained after trimming. The microbiota of all fecal samples after QIIME2 was composed of 2606 OTUS with a relative abundance higher than 1% in at least two samples (97% similarity cut-off).

The Chao1 index (community richness) and Shannon index (microbiota diversity) were calculated at genus level to estimate the alpha diversity of the components of the fecal microbiota in the study groups. The Chao1 index values for each group suggested a significant decrease in richness in both CRC groups compared to healthy controls (L-CRC vs. L-HC, *p* < 0.001; OB-CRC vs. L-HC, *p* = 0.035) (Figure 1A). Moreover, a significant decrease in Shannon diversity was found in L-CRC and OB-CRC patients compared to L-HC controls (L-CRC vs. L-HC, *p* = 0.0014; OB-CRC vs. L-HC, *p* = 0.039) (Figure 1B). The alpha diversity comparison revealed no different levels of diversity and richness between L-CRC and OB-CRC patients (Shannon: L-CRC vs. OB-CRC, *p* = 0.34; Chao1: L-CRC vs. OB-CRC, *p* = 0.37) (Figure 1C).

On the other hand, differences in microbiota communities (β-diversity) between study groups were determined by calculating the Bray–Curtis dissimilarity index. The ordination plots showed a significant separation in the bacterial communities in both L-CRC and OB-CRC patients with respect to L-HC controls (L-CRC vs. L-HC, *p* = 0.004; OB-CRC vs. L-HC, *p* = 0.007, ANOSIM) (Figure 2A,B). Again, no significant differences in beta diversity were found between L-CRC and OB-CRC groups (*p* = 0.485, ANOSIM) (Figure 2C).

### 2.2. Taxonomy of Fecal Microbiota in the Study Groups

The analysis of the distribution of the fecal microbiota at the phylum level indicated that Bacteroides and Firmicutes were the predominant phyla in the three study groups. Other phyla such as Proteobacteria, Actinobacteria, Fusobacteria, and Lentisphaerae were also relatively abundant in all groups, while Tenericutes, Synergistetes Verrumicrobia, Spirochaetes, and WS6 were detected a low relative abundance (<1%). Comparison of the abundance of these phyla between study groups, revealed a significant increase in the abundance of Firmicutes (L-CRC vs. L-HC, q < 0.001; OB-CRC vs. L-HC, q = 0.005), Fusobacteria (L-CRC vs. L-HC, q = 0.002; OB-LCR vs. L-HC, q = 0.001), and Proteobacteria (L-CRC vs. L-HC, q = 0.012; OB-CRC vs. L-HC, q = 0.014), and a significant decrease in the abundance of Bacteroidetes (L-CRC vs. L-HC, q < 0.001; OB-CRC vs. L-HC, q = 0.005) in the CRC groups (non-obese and obese) compared to the non-obese healthy controls. Additionally, we found a significantly higher abundance of the phylum Lentisphaerae in L-CRC subjects compared to L-HC controls (q = 0.010). Significantly higher levels of Firmicutes (q = 0.008) and Proteobacteria (q = 0.013) were also found in the OB-CRC group with respect to the L-CRC group (Figure 3).

Twenty-seven bacterial families were detected in all study patients. Both CRC groups (L-CRC and OB-CRC) displayed a significantly higher abundance of the Fusobacteriaceae (L-CRC vs. L-HC, q = 0.002; OB-CRC vs. L-HC, q = 0.001), Prevotellacea (L-CRC vs. L-HC, q = 0.010; OB-CRC vs. L-HC, q = 0.003), Clostridiaceae (L-CRC vs. L-HC, q = 0.019; OB-CRC vs. L-HC, q = 0.020), Barnesiellaceae (L-CRC vs. L-HC, q = 0.002; OB-CRC vs. L-HC, q = 0.025), Porphyromonadaceae (L-CRC vs. L-HC, q = 0.010; OB-CRC vs. L-HC, q = 0.035), and Desulfovibrionaceae (L-CRC vs. L-HC, q = 0.005; OB-CRC vs. L-HC, q = 0.003) when compared to L-HC controls. Furthermore, a significantly lower abundance of Ruminoccocacea (L-CRC vs. L-HC, q = 0.018; OB-CRC vs. L-HC, q = 0.001) and Bacteroidaceae (L-CRC vs. L-HC, q = 0.002; OB-CRC vs. L-HC, q = 0.010) were found in the CRC groups with respect to the L-HC group. Victivallaceae was also significantly enriched in L-CRC patients compared to L-HC subjects (q = 0.010), while Enterobacteraceae (OB-CRC vs. L-CRC, q = 0.040; OB-CRC vs. L-HC, q = 0.029) and Streptococcaceae (OB-CRC vs. L-CRC, q = 0.004; OB-CRC vs. L-HC, q = 0.016) were significantly increased in the OB-CRC group compared to L-CRC and L-HC groups (Figure 4).

Further analysis revealed significant differences in the microbial composition at the genus level between the study groups. A total of 39 genera were identified among the 60 fecal samples, with only significant differences in 14 genera between CRC patients and healthy individuals. Thus, the genera *Prevotella* (L-CRC vs. L-HC, q = 0.001; OB-CRC vs. L-HC, q = 0.003), *Clostridium* (L-CRC vs. L-HC, q = 0.019; OB-CRC vs. L-HC, q = 0.030), *Desulfovibrio* (L-CRC vs. L-HC, q = 0.002; OB-CRC vs. L-HC, q = 0.006) and *Enterococcus* (L-CRC vs. L-HC, q = 0.031; OB-CRC vs. L-HC, q = 0.05) were significantly increased in both CRC groups compared to the L-HC group. In addition, the relative abundance of *Bacteroides* (L-CRC vs. L-HC, q = 0.003; OB-CRC vs. L-HC, q = 0.045), *Butyricimonas* (L-CRC vs. L-HC, q = 0.001; OB-CRC vs. L-HC, q = 0.012), *Roseburia* (L-CRC vs. L-HC, q = 0.021; OB-CRC vs. L-HC, q = 0.019), *Ruminococcus* (L-CRC vs. L-HC, q = 0.018; OB-CRC vs. L-HC, q = 0.035), and *Alistipes* (L-CRC vs. L-HC, q = 0.005; OB-CRC vs. L-HC, q = 0.037) were significantly decreased in both CRC groups with respect to the L-HC group. Finally, other genera such as *Victivallis* was significantly elevated in the L-CRC compared to L-HC controls (q = 0.012). In OB-CRC patients we found that *Enterobacter* (OB-CRC vs. L-CRC, q = 0.038; OB-CRC vs. L-HC, q = 0.002), *Escherichia* (OB-CRC vs. L-CRC, q = 0.024; OB-CRC vs. L-HC, q = 0.006), *Fusobacterium* (OB-LCR vs. L-CRC, q = 0.003; OB-CRC vs. L-HC, q = 0.002), and *Streptococcus* (OB-CRC vs. L-CRC, q = 0.038; OB-CRC vs. L-HC, q = 0.05) were significantly enriched, while the relative abundance of *Blautia* (OB-CRC vs. L-CRC, q = 0.012; OB-CRC vs. L-HC, q = 0.019) and *Faecalibacterium* (OB-CRC vs. L-CRC, q = 0.030, OB-CRC vs. L-HC, q = 0.024) were significantly lower when compared to L-CRC and L-HC individuals (Figure 5).

At the species levels, we found a significant rise in the abundance of *Enterococcus faecalis* (L-CRC vs. L-HC, q = 0.004; OB-CRC vs. L-HC, q = 0.012), and a significant decline in the abundance of *Bacteroides caccae* (L-CRC vs. L-HC, q = 0.007; OB-CRC vs. L-HC, q = 0.029), *Ruminoccocus lactaris* (L-CRC vs. L-HC, q = 0.05; OB-CRC vs. L-HC, q = 0.019), *Alistipes putredinis* (L-CRC vs. L-HC, q = 0.011; OB-CRC vs. L-HC, q = 0.05), and *Alistipes indistinctus* (L-CRC vs. L-HC, q = 0.008; OB-CRC vs. L-HC, q = 0.010), in both L-CRC and OB-CRC patients in comparison to L-HC controls. *Victivallis vadensis* (q = 0.012) was significantly higher and *Bacteroides uniformis* (q = 0.010) was significantly lower in L-CRC patients compared to healthy controls. Finally, *Clostridium septicum* (OB-CRC vs. L-CRC, q = 0.025; OB-CRC vs. L-HC, q = 0.004), *Escherichia coli* (OB-CRC vs. L-CRC, q = 0.027; OB-CRC vs. L-HC, q = 0.007), *Fusobacterium nucleatum* (OB-CRC vs. L-CRC, q = 0.003; OB-CRC vs. L-HC, q = 0.001), *Enterobacter cloacae* (OB-CRC vs. L-CRC, q = 0.013; OB-CRC vs. L-HC, q = 0.009), and *Streptoccoccus bovis* (OB-CRC vs. L-CRC, q = 0.011; OB-CRC vs. L-HC, q = 0.027) were significantly enriched, while *Faecalibacterium prausnitzii* (OB-CRC vs. L-CRC q = 0.011; OB-CRC vs. L-HC q = 0.043) was significantly reduced in OB-CRC patients compared to L-CRC and L-HC subjects.

### 2.3. Serum Zonulin Levels

Serum zonulin levels were significantly higher in the OB-CRC group compared to L-HC (26.57 ± 14.95 vs. 14.72 ± 9.57, *p* < 0.001) and L-CRC groups (26.57 ± 14.95 vs. 20.07 ± 15.23, *p* = 0.013). Furthermore, the zonulin levels showed a non-significant trend towards increased concentrations in L-CRC patients compared to L-HC controls (20.07 ± 15.23 vs. 14.72 ± 9.57, *p* = 0.804).

### 2.4. Relationship between the Fecal Microbiota and Serum Levels of Zonulin, TMAO, and Inflammatory Mediators in the Study Groups

Correlation analyses between the abundance of specific bacteria at different taxa levels and serum levels of zonulin, TMAO, and inflammatory mediators (IL-1β and IL-10) in all study groups are shown in Table 2 and Table 3.

Subsequent lineal regression analysis showed that the relative abundances of *Ruminococcus* (R^2^ = 0.33, β = −0.554, *p* = 0.014) and *Blautia* (R^2^ = 0.33, β = −0.925, *p* = 0.024) were negatively associated with zonulin levels in the L-HC control group. Nevertheless, the abundance of *Prevotella* (R^2^ = 0.33, β = 0.978, *p* = 0.003) was positively associated with serum zonulin level in OB-CRC patients, while abundance of *Desulfovibrio* (R^2^ = 0.33, β = 0.787, *p* = 0.014) was positively associated to serum zonulin levels in L-CRC patients.

Similarly, regression analysis showed that, in the case of L-HC control subjects, the levels of the anti-inflammatory factor IL-10 were positively associated with the abundance of *Roseburia* (R^2^ = 0.38, B = 0.681, *p* = 0.001), while the levels of the inflammatory factor IL-1β were positively associated with the abundance of *Enterobacter* (R^2^ = 0.38, B = 0.435, *p* = 0.048). In addition, the level of IL-1β in OB-CRC patients was positively associated with the abundance of *Fusobacterium nucleatum* (R^2^ = 0.38, B = 0.1963, *p* = 0.050), while the level of IL-10 was positively associated with the abundance of *Blautia* (R^2^ = 0.38, B = 0.555, *p* = 0.009) and *Faecalibacterium prausnitzii* (R^2^ = 0.38, B = 0.456, *p* = 0.026). Finally, in the L-CRC group the levels of IL-1β and IL-10 were positively associated with the abundance of *Enterococcus faecalis* (R^2^ = 0.41, B = 0.418, *p* = 0.037) and the abundance of *Ruminococcus* (R^2^ = 0.41, B = 0.418, *p* = 0.022), respectively.

On the other hand, serum levels of TMAO were found to be positively associated with the abundance of *Enterobacteriaceae* (R^2^ = 0.43, B = 0.618, *p* = 0.005) and *Escherichia coli* (R^2^ = 0.43, B = 0.812, *p* = 0.012) in OB-CRC patients, and with the abundance of *Desulfovibrio* (R^2^ = 0.52, B = 0.576, *p* = 0.003) in L-CRC patients. No significant associations were found between any bacterial group and the serum TMAO levels in L-HC controls.

### 2.5. Predicted Functional Metagenome Analysis

Phylogenetic Investigation of Communities by Reconstruction of Unobserved States (PICRUSt) was used to identify differences in metagenome functional prediction based on Greengenes 16S rRNA database and KEGG Orthologs. The PICRUSt analysis showed that genes involved in energy metabolism (oxidative phosphorylation, q = 0.022), methane and sulfur metabolism (q = 0.015 and q = 0.023, respectively), and glycan biosynthesis and metabolism (lipopolysaccharide biosynthesis, q = 0.05; glycosiltransferases, q = 0.043) were significantly over-represented according to the BMI increase when CRC groups and the L-HC group were compared. Moreover, carbohydrate metabolism (Citrate cycle (TCA cycle), q = 0.032), butanoate metabolism, q = 0.035, and pentose phosphate pathway, q = 0.016), amino acid metabolism (glycine, serine, and threonine metabolism, q = 0.005; valine, leucine, and isoleucine biosynthesis, q = 0.005), metabolism of other amino acids (selenocompound metabolism, q = 0.006) and protein processing in endoplasmic reticulum (q = 0.047) were over-represented in the L-CRC group when compared to the OB-CRC group (Figure 6).

Finally, when comparing L-HC controls to both CRC groups we found that gut microbiota from CRC patients was significantly enriched with genes implicated in antigen processing and presentation (*p* = 0.004), bacterial chemotaxis (q = 0.013), bacterial secretion system (q = 0.007), and bacterial toxin (q = 0.007) and significantly reduced in genes related to ABC transporters (q = 0.022), xenobiotic degradation and metabolism (ethylbenzene degradation q = 0.026) and lipid metabolism functions (fatty acid biosynthesis and degradation (q = 0.017), glycerophopholipid metabolism (q = 0.034), and arachidonic acid metabolism (q = 0.04)) (Figure 6).

## 3. Discussion

In this study we showed that the composition of the gut microbiome from CRC patients (both obese and non-obese) was significantly different to the gut eubiotic microbiota of non-obese healthy subjects. Moreover, we found an obesity-related microbial profile linked to CRC, that could be responsible for the significantly higher serum levels of zonulin (marker of intestinal permeability), TMAO (CVD-related microbial metabolite), and IL-1β (proinflammatory factor) and the lower levels of IL-10 (anti-inflammatory factor) compared to non-obese CRC patients and controls.

The analysis of the alpha diversity (community composition) of the gut microbiome from the three groups revealed a decreased richness (Chao1 index) and diversity (Shannon index) in the OB-CRC and L-CRC groups compared to L-HC controls. Nevertheless, no significant differences in Chao1 and Shannon indices were found between OB-CRC and L-CRC groups. These results may suggest that the decrease in gut microbiota diversity of CRC patients could not be entirely related to a history of obesity. In addition, the Bray–Curtis dissimilarity plot analysis to detect microbial community differences in structure clustered OBC-CRC and L-CRC patients together, but clustered L-HC controls separately, confirming that obesity does not introduce important changes to the overall structure of the gut microbial community in CRC patients. In this regard, a recent meta-analysis done by Greathouse et al. described no universal differences in alpha and beta diversity between obese and non-obese patients with CRC, suggesting that, similarly to the community composition, community structure is not associated with BMI in CRC patients [23].

Furthermore, the present study demonstrated that CRC patients exhibit clear differences in gut microbiota composition when compared to healthy individuals, independently of the BMI of the patient. On one hand we identified an increase of Firmicutes, Fusobacteria, and Proteobacteria phyla in fecal samples from CRC patients. Remarkably, these phyla have been previously associated with gut dysbiosis, inflammation, and CRC [24]. On the other hand, genus-level analyses confirmed that the intestinal microbiota of CRC patients with or without obesity is characterized by a reduction of butyrate-producing bacteria (*Butyricimonas*, *Roseburia*, *Blautia, Faecalibacterium*, and *Ruminococcus*) and an increase in harmful bacterial species that could act as opportunist pathogens with pro-inflammatory and pro-carcinogenic properties (*Fusobacterium, Clostridium, Prevotella, Desulfovibrio*, and *Enterococcus*). Accordingly, other works have shown that CRC patients display an enrichment in pro-inflammatory opportunistic pathogens and a decrease in butyrate-producing bacteria, which may lead to an imbalance in intestinal homeostasis (dysbiosis) that could ultimately lead to tumor formation [25,26]. These CRC-related significant alterations in specific bacterial groups have been proposed to have a potential impact on mucosal immune response [27].

Nevertheless, we found a significant increase in the abundance of several specific taxa of opportunist pathogens in the gut microbiome of OB-CRC patients compared to L-CRC and L-HC subjects. In particular, in the obese CRC group we detected a significant rise in the abundance of the families Enterobacteraceae and Streptococcaceae and the genera/species *Enterobacter* (*E. cloacae*), *Escherichia* (*E. coli*), and *Streptococcus* (*S. bovis*). Enterobacteriaceae are normal commensal bacteria in the human gut. However, the family includes numerous genera of bacteria that are potentially pathogenic, such as *Salmonella*, *Shigella*, *Escherichia*, *Enterobacter*, *Proteus*, and *Klebsiella* [28]. Previous studies have reported that Enterobacteriaceae is more abundant in patients with inflammatory bowel disease (IBD) or CRC in comparison to healthy individuals [29].

Gut microbiota might directly influence the relationship between obesity and CRC. In this study, we found that OB-CRC patients have significantly higher plasma levels of TMAO when compared with L-CRC and L-HC subjects. Barrea et al. demonstrated that circulating levels of TMAO increased along with BMI in patients with overweight or obesity [30]. Another recent study also reported increased serum TMAO levels among CRC patients, compared to healthy controls, rendering TMAO as a potential prognostic marker for CRC [31]. Additionally, we found that the presence of certain specific bacterial taxa in human feces of both CRC groups were associated with the concentration of plasma TMAO. We observed that the serum TMAO concentrations were significantly and positively associated with the abundance of the family *Enterobacteriaceae* and species *Escherichia coli* in OB-CRC patients and the abundance of *Desulfovibrio* in L-CRC patients. In agreement with our results, other human and animal studies have suggested that several families of bacteria are involved in the production of TMA/TMAO such as Prevotellaceae [32] and Enterobacteriaceae [33,34]. Moreover, a novel microbial, the cntA/B, has been found to be able to convert carnitine into TMA and this gene was reported to exist among only few species including *Escherichia coli, Klebsiella* spp., and *Citrobacter* spp. [35]. Additionally, it has been previously described that the increase in the conversion of choline to TMA can be caused also by the expression of the cutC gene by bacteria such as *Desulfovibrio* [33]. Then, the increase of specific pathogenic bacteria such as *Escherichia coli* in OB-CRC patients can be responsible (at least partially) for the significant increase in microbial-derived proinflammatory molecules such as TMAO.

Nevertheless, blood TMAO levels not only depend on the gut microbiota composition and metabolic activities [20], but also on the functioning of the gut–blood barrier that controls the access of gut-derived molecules to the bloodstream [36]. Accordingly, we found that plasma zonulin levels were significantly higher in the OB-CRC patients compared to L-CRC and L-HC controls. Increased zonulin level was associated with the abundance of *Prevotella* in OB-CRC patients. *Prevotella* contains enzymes that are important in mucin degradation, which may disrupt the colonic mucus barrier and impair the intestinal barrier function [37], and therefore may contribute to increase the circulating levels of TMAO. Recent evidence has suggested that TMAO could play a role in the inflammatory process and that this inflammation induction could be a possible factor that provides a link between TMAO and cancer. Serum level of TMAO was found to synergize the pro-inflammatory effects of *Helicobacter pylori* infections on gastric epithelial cells, through the increase in the expression level of pro-inflammatory genes such as IL-6 and chemokine ligands [38], which play roles in cancer progression [39]. In another study, Yue et al. also demonstrated that TMAO can trigger the activation of the NOD-like receptor family pyrin domain containing 3 (NLRP3) inflammasome [40], which has been suggested to be implicated in the growth and/or metastasis of a variety of cancers including CRC [41]. Nevertheless, further research is necessary to specify the mechanism by which TMAO is linked to CRC via inflammation induction.

With respect to the obesity-specific microbiota observed in in OB-CRC patients, the passenger *Fusobacterium nucleatum* has been reported to be more abundant in people who are obese than in lean people [42]. *Fusobacterium nucleatum* is an opportunistic pathogen closely associated with the occurrence and development of periodontitis, whose relationship with CRC has been widely reported [43,44,45]. We found an association of this species with the higher levels of the proinflammatory IL-1β in OB-CRC patients. Increased abundance of *Fusobacterium nucleatum* has been previously associated to a higher expression of pro-inflammatory cytokines in colonic tissue from CRC patients [46,47]. Thus, Kostic et al. suggested that *Fusobacterium nucleatum* induced a nuclear factor-κB-driven proinflammatory response to promote CRC [48]. In addition, Rubinstein et al. described that *Fusobacterium* spp. function includes the induction of inflammatory responses and colon tumor cell growth promotion via β-catenin activation [49]. Furthermore, IL-1β induces the activation of the Wnt signaling pathway by phosphorylation of GSK3β [50], a key signaling pathway for intestinal tumorigenesis [51], supporting the central role of IL-1β in CRC pathogenesis.

On the other hand, previous studies have described that IL-10 deficiency leads to elevated levels of TNF-alpha, IL-6, and IL-17, triggering chronic inflammation and promoting tumor growth [52]. In this study, the lowest levels of anti-inflammatory IL-10 found in the OB-CRC patients were associated to the lowest abundance of *Blautia* and *Faecalibacterium prausnitzii.* All these bacteria are important suppliers of butyrate to the colonic epithelium. Butyrate is a short chain fatty acid (SCFA) considered as the most important nutrient for epithelial cells of the colon and has a role in the epigenetic control of gene expression, while also functioning as a mediator of anti-inflammatory responses, the maintenance of the intestinal barrier integrity, and the protection against oxidative stress [53,54]. Therefore, butyrate promotes the integrity of gut epithelial tight junctions as well as increases the release of the anti-inflammatory cytokine IL-10 [55], that protects against cancer formation.

Finally, our Picrust analysis suggests a lower relative abundance of genes responsible for carbohydrate metabolism functions such as butanoate metabolism and pentose phosphate pathway, together with genes for the amino acid metabolism and protein processing in endoplasmic reticulum were found depleted in OB-CRC patients compared to L-CRC patients. The relative abundance of genes of the pentose phosphate pathway is critical for cancer cells due to the generation of high levels of NADPH, which may be utilized for the nucleic and fatty acids synthesis and in the cell survival under stress conditions [56]. Moreover, a significant over-representation of genes for energy metabolism such as oxidative phosphorylation, methane metabolism, and sulfur metabolism as well as for lipopolysaccharide biosynthesis were found increase in OB-CRC patients with respect to L-CRC patients and L-HC controls. Sulfur-metabolizing microbes, which convert dietary sources of sulfur into genotoxic hydrogen sulfide (H_2_S), have been previously associated with development of CRC [57]. Moreover, gut-derived H_2_S may fragment the mucus bilayer of the gastrointestinal tract and this breach may precede tumorigenesis by exposing gut epithelium to immunogenic luminal bacteria [58]. The significant increase of genes for lipopolysaccharide biosynthesis found in the OB-CRC groups could be in part attributed to the significant increase in the abundance of *Escherichia coli* and other species of the family Enterobacteriaceae, which contain specific enzymes to produce LPS [59]. These results suggest that the microbial differences observed in OB-CRC patients could be associated with changes in functional pathways.

## 4. Materials and Methods

### 4.1. Study Subjects

Forty-five patients aged 35–75 years with stages II–III (T2-T4 and/or N1-N2) were recruited at the Radiotherapy Oncology Service at Virgen de la Victoria Hospital. Patients were enrolled at initial diagnosis and did not receive any treatment before collection of fecal and peripheral blood samples. Patients were dichotomized into non-obese (BMI < 30 kg/m^2^) (L-CRC) and obese (BMI ≥ 30 kg/m^2^) (OB-CRC) according to the WHO guidelines. Exclusion criteria were familial colorectal cancer, presence of inflammatory bowel disease, food allergies, use of antibiotics within the past 3 months before sampling, or regular use of non-steroidal anti-inflammatory drugs, statins, or probiotics.

Additionally, 20 non-obese healthy controls (L-HC) (BMI < 30 kg/m^2^) (age- and gender-matched controls) were recruited for the study. The exclusion criteria for healthy controls included gut disease diagnosis and/or medication, and previous CRC diagnosis.

The study protocol was approved by the Clinical Research Ethics Committee at the Virgen de la Victoria University Hospital and conducted in accordance with the Declaration of Helsinki. All participants provided written informed consent.

### 4.2. Laboratory Measurements

Blood samples were collected from study patients after an overnight fast of at least 12 h. Serum was separated by centrifugation and aliquots were immediately frozen at −80 °C. Levels of fasting glucose, total cholesterol, triglycerides, HDL-cholesterol, LDL-cholesterol, and glycated hemoglobin (HbA1c) were measured in duplicates using a Dimension autoanalyzer (Dade Behring Inc., Deerfield, IL, USA) by enzymatic methods (Randox Laboratories Ltd., Crumlin, UK).

### 4.3. DNA Extraction and Gut Microbiota Sequencing

DNA extraction was performed from 200 mg of fecal material using the QIAamp DNA stool Mini kit (Qiagen, Hilden, Germany) following the manufacturer’s instructions. DNA concentration was determined by absorbance at 260 nm (A260), and purity was estimated by determining the A260/A280 ratio with a Nanodrop spectrophotometer (Nanodrop Technologies, Wilmington, DE, USA).

DNA was amplified using the 16S Metagenomics kit (Thermofisher, Waltham, MA, USA) that contains a primer pools to amplify multiple variable regions (V2, 3, 4, and 6–9) of the 16S rRNA. After, the Ion PlusTM Fragment Library Kit (Thermofisher) was used to ligate barcoded adapters to the generated amplicons and create the barcoded libraries, which were pooled and template on the automated Ion Chef system (Thermofisher). The sequencing was done on the Ion S5 platform (Thermofisher).

### 4.4. Bioinformatics Analysis

Analysis of 16S rRNA amplicons was done using QIIME 2-2019.4 [60]. Raw sequence data were quality filtered and denoised, dereplicated, and chimera filtered using the q-dada2 plugin with DADA2 pipeline. q2-feature-table plugin was used to merge into a single feature table the sequence variants obtained by the DADA2 pipeline. All amplicon sequence variants from the merged feature table were clustered into OTU’s with Open Reference Clustering method against the Greengenes version 13_8 with 97% of similarity from OTUs reference sequences using the q2-vsearch plugin with 97% similarity of sequence. The OTUs were aligned with MAFFT (via q2-alignment) and used to construct a phylogeny with fasttree2 (via q2-phylogeny). Taxonomy was assigned to OTUs using the q2-feature-classifier classify-sklearn naïve Bayes taxonomy classifier. Alpha diversity metrics (Shannon and Chao1), beta diversity metrics (Bray–Curtis dissimilarity), and Principle Coordinate Analysis (PCoA) were estimated using q2-diversity after samples were rarefied to 994 sequences per sample. Alpha diversity significance was estimated with Kruskal–Wallis test using q2-diversity plugin. Beta diversity significance was estimated using ANOSIM statistical method.

The PICRUSt analysis was used to predict metagenome function by picking OTUs against the Greengenes database [61]. The R packages “pheatmap” were used for data analysis and plotting. Statistical analysis was done in R 3.6.0. *p*-values were corrected for multiple comparisons using the Benjamini–Hochberg method. A corrected *p* < 0.05 was considered as statistically significant.

### 4.5. Intestinal Permeability Analysis

Serum level of zonulin was determined by commercially available enzyme-linked immunosorbent (ELISA) assays (Immundiagnostik AG, Bensheim, Germany) according to the manufacturers’ protocol. The detection limit of the assay was 0.22 ng/mL, whereas the intra and inter-assay coefficients of variation ranged from 3% to 10%. Standards and study samples were tested in duplicate.

### 4.6. Cytokine Analysis

Serum levels of IL-10 and IL-1β were measured by ELISA assays (Novex, Life Technology). Detection ranges were 7.8–500 and 3.9–250 pg/mL for IL-10 and IL-1β, respectively.

### 4.7. Quantification Trimethylamine N-oxide (TMAO) in Serum Samples

TMAO concentrations were quantified in serum samples using Nuclear Magnetic Resonance (NMR) as previously described [62]. The NMR spectra were analyzed in a Bruker Avance IVDr 600 MHz spectrometer (Bruker Biospin, Rheinstetten, Germany) and processed using the TopSpin program (Bruker Biospin, Germany). Three NMR spectra were analyzed separately for each sample. The intensity of the TMAO peaks were measured for all spectra and the concentrations in each sample were calculated with fitted calibration curves.

### 4.8. Statistical Analyses

Wilcoxon rank-sum test and Welch’s *t*-test were used to compare the bacterial abundance between study groups and false discovery rate (FDR) using the Benjamini–Hochberg method was applied to correct the significant *p*-values (q < 0.05).

Differences in the clinical and biochemical variables between the three study groups were analyzed by Kruskal–Wallis test and subsequent post hoc Bonferroni and the differences between two groups were analyzed using Mann–Whitney U test. Linear correlations between variables were calculated by the Spearman correlation coefficient. A linear regression analysis was done to identify what bacteria was an independent predictor for serum inflammatory mediators, TMAO, and zonulin levels in each study group. Statistical analysis was performed with SPSS version 26.0 (SPSS Inc., Chicago, IL, USA). The level of significance was set at *p* < 0.05 for all analyses.

## 5. Conclusions

This study established an association between inflammation, BMI, and gut microbiota in CRC patients. Firstly, we showed that obesity does not induce significant changes in the diversity and richness of intestinal bacteria of CRC patients. Secondly, we demonstrated that the presence of obesity in CRC patients is associated to changes in the composition and functionality of the gut microbiota. Thus, the gut microbiota of CRC patients with obesity is characterized by the presence of a higher abundance of opportunistic pathogens (such as *Prevotella, Fusobacterium nucleatum*, Enterobacteriaceae, and *Escherichia coli*), which may impair intestinal barrier function (increased circulating zonulin levels), and may contribute to inflammatory processes related to CRC by means of increasing the production of inflammatory molecules such as IL-1β and TMAO. Although it is possible that our study has some limitations in the statistical analysis because of the multiple testing and it should be replicated in larger cohorts (including other populations with different eating patterns and cultural food), overall our results strongly suggest an important role of gut microbiota in the development of CRC in patients with obesity. Furthermore, these finding could provide new clues for the development of diagnostic tools for CRC prevention.

## Figures and Tables

**Figure 1 ijms-21-06782-f001:**
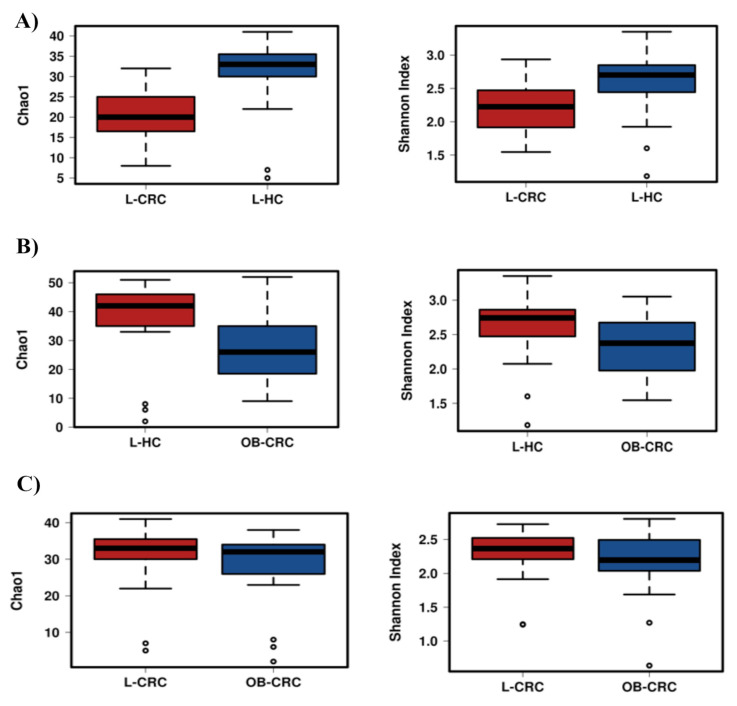
Richness (Chao1 index) and diversity (Shannon index) indices between microbial communities from feces of CRC patients with obesity (OB-CRC), non-obese CRC patients (L-CRC), and non-obese healthy controls (L-HC) at the genus level. (**A**) L-CRC vs. L-HC, (**B**) L-HC vs. OB-CRC, and (**C**) L-CRC vs. OB-CRC.

**Figure 2 ijms-21-06782-f002:**
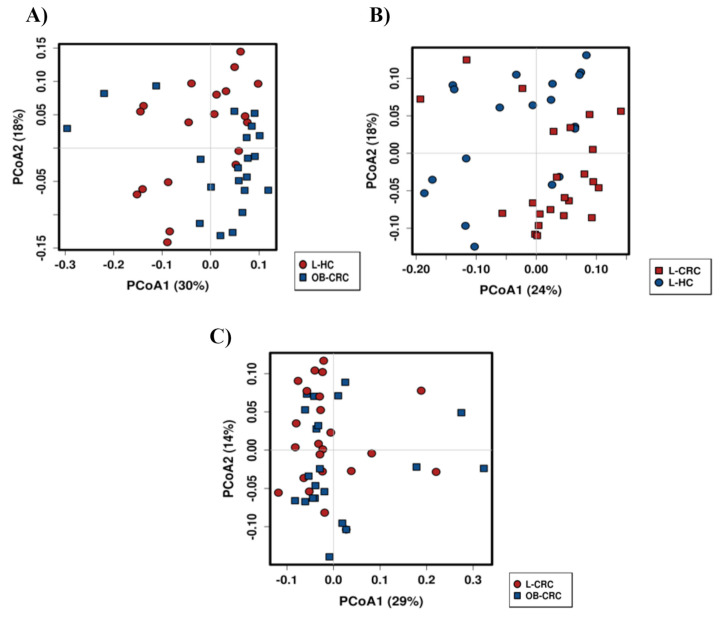
Clustering of fecal bacterial communities according to the different study groups by PCoA using Bray–Curtis dissimilarity matrix. Each point corresponds to a community coded according to the study groups. The percentage of variation explained by the plotted principal coordinates is indicated on the axes. (**A**) L-HC vs. OB-CRC, (**B**) L-CRC vs. L-HC, and (**C**) L-CRC vs. OB-CRC.

**Figure 3 ijms-21-06782-f003:**
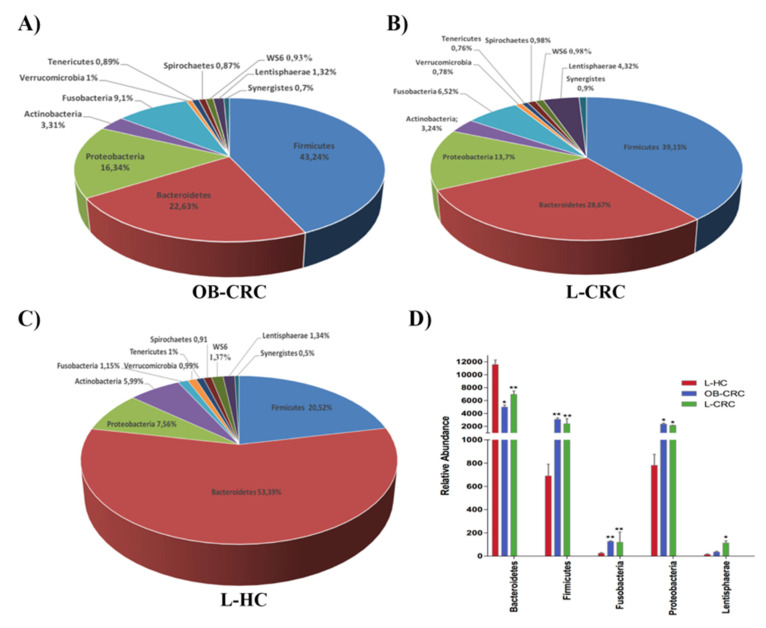
Phylum-level distributions of bacteria in fecal samples of (**A**) OB-CRC, (**B**) L-CRC, and (**C**) L-HC groups. Data are shown as a percentage of the total identified sequences per group. (**D**) Differentially abundant phyla in the stool samples of OB-CRC and L-CRC patients compared to L-HC * *p* < 0.05, ** *p* < 0.001. The bars indicate mean ± SD.

**Figure 4 ijms-21-06782-f004:**
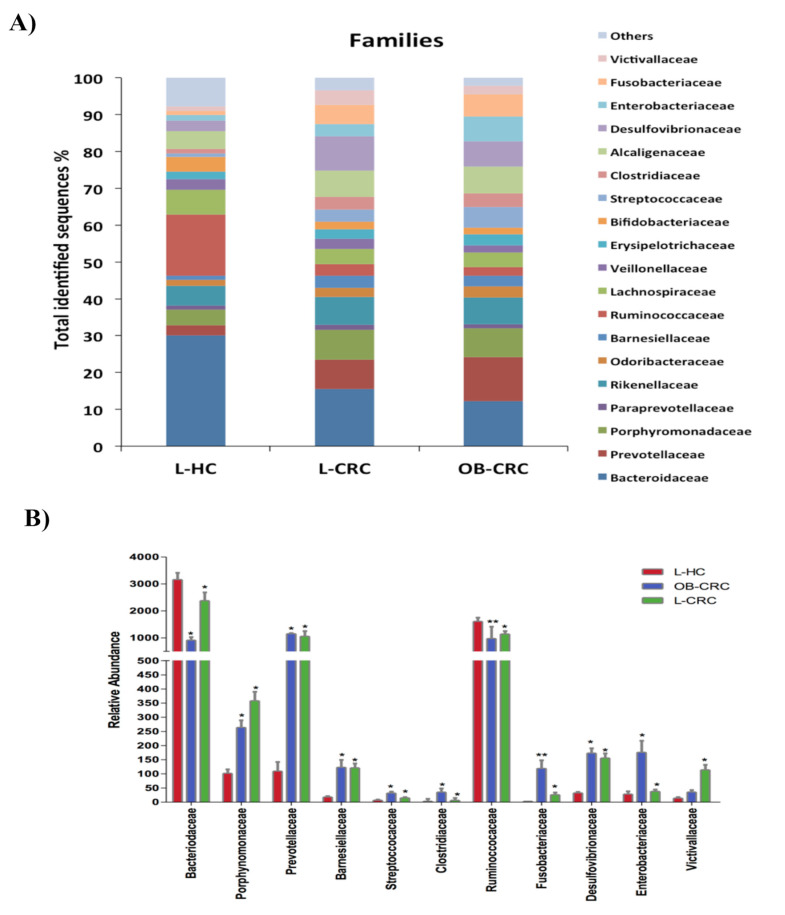
Family-level microbial classification of bacteria from OB-CRC, L-CRC, and L-HC stool samples. (**A**) Data are shown as a percentage of the total identified sequences per group. (**B**) Differentially abundant families in the stool samples of OB-CRC and L-CRC patients compared to L-HC * *p* < 0.05, ** *p* < 0.001. The bars indicate mean ± SD.

**Figure 5 ijms-21-06782-f005:**
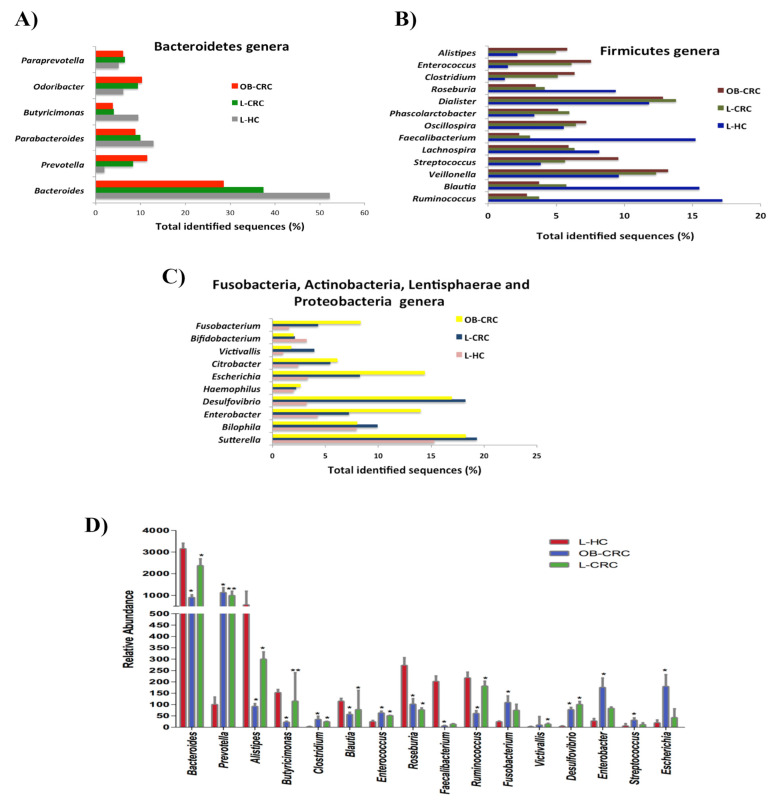
Relative abundance of bacterial genera in the microbiota of OB-CRC, L-CRC, and L-HC controls. (**A**) Bacteroidetes genera, (**B**) Firmicutes genera, (**C**) Fusabacteria, Actinobacteria, Lentisphaerae, and Proteobacteria genera. Data are shown as a percentage of the total identified sequences per group. (**D**) Differentially abundant genera in the stool samples of OB-CRC and L-CRC patients compared to L-HC. * *p* < 0.05, ** *p* < 0.001. The bars indicate mean ± SD.

**Figure 6 ijms-21-06782-f006:**
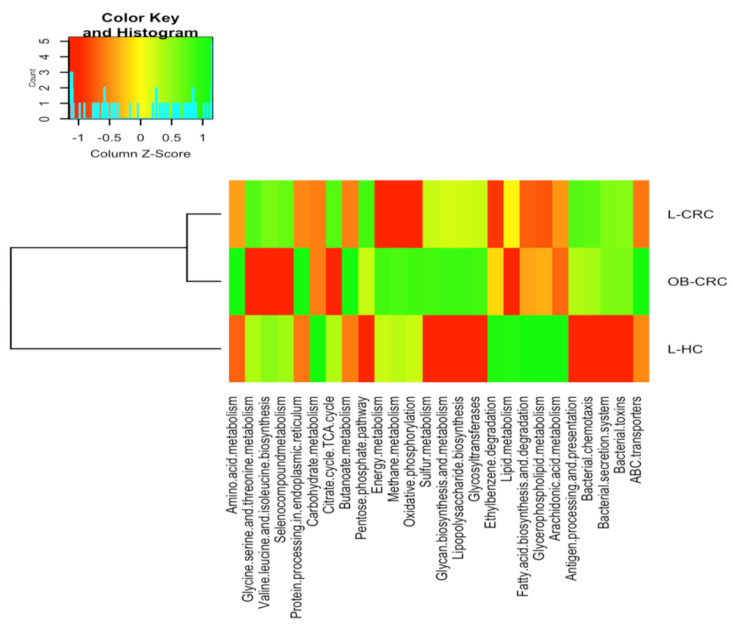
Predicted functional composition of metagenomes based on 16S rRNA gene sequencing data of OB-CRC, L-CRC, and L-HC controls. Heatmap of differentially abundant KEGG pathways identified in the three study groups. The values of color in the heatmap represent the normalized relative abundance of KEGG pathways.

**Table 1 ijms-21-06782-t001:** Clinical and biochemical characteristics, serum levels of trimethylamine N-oxide (TMAO) and inflammatory markers in the study groups.

	L-HC N = 20	L-CRC N = 23	OB-CRC N = 22	*p*
Age (years)	61.42 ± 7.40	62.52 ± 7.99	64.43 ± 7.31	0.208
Gender, n (M/F)	10/10	12/11	11/11	
BMI (kg/m^2^)	25.45 ± 3.23 ^a^	25.32 ± 3.67 ^a^	35.82 ± 3.83 ^b^	0.001
Constipation, n (%)	6 (20%)	8 (26.6%)	10 (33.3%)	0.383
Alcohol consumption, n (%)	4 (13.3%)	4 (13.3%)	3 (10%)	0.997
Current smoking, n (%)	9 (30%)	12 (40%)	10 (33.3%)	0.588
**Biochemical data**				
Glucose (mg/dL)	94.85 ± 9.86	92.04 ± 10.91	108.42 ± 10.53	0.456
Total cholesterol (mg/dL)	175.2 ± 33.6	187.12 ± 20.74	193.09 ± 19.91	0.325
Triglycerides (mg/dL)	112.67 ± 34.51	110.32 ± 33.03	127.7 ± 22.6	0.510
HDL-cholesterol (mg/dL)	60.7 ± 15.1	54.84 ± 18.41	47.28 ± 15.6	0.755
LDL-cholesterol (mg/dL)	107.78 ± 27.12	109.84 ± 25.98	112.80 ± 25.34	0.678
IL-1Β (pg/mL)	76.40 ± 9.81 ^a^	103.32 ± 9.43 ^b^	110.65 ± 12.98 ^c^	0.006
IL-10 (pg/mL)	155.19 ± 9.60 ^a^	121.96 ± 15.22 ^b^	102.21 ± 9.82 ^c^	0.004
TMAO (ng/mL)	12.72 ± 9.57 ^a^	20.07 ± 15.23 ^b^	26.57 ± 14.95 ^c^	0.003
**Histological variables**				
Stages				
II		10 (43.47%)	12 (54.54%)	0.998
III		13 (56.52%)	10 (45.45%)	0.997
Tumor depthpenetration (T)				
T2–T3		14 (60.86%)	15 (68.18%)	0.775
T4		9 (39.13%)	7 (31.81%)	0.768
Grade of differentiation				
G1		9 (39.13%)	10 (45.45%)	0.995
G2		6 (26.08%)	5 (22.72%)	0.438
G3		4 (17.39%)	5 (22.72%)	0.998
No differentiation		3 (13.04%)	2 (9.09%)	0.997

BMI: body mass index; HDL: high density lipoprotein; LDL: low density lipoprotein TMAO: trimethylamine N-oxide; IL-1β: interleukin-1 beta; IL-10: interleukin 10. Values are expressed as mean ± SD. Different superscript letters indicate significant differences between study groups *p* < 0.05.

**Table 2 ijms-21-06782-t002:** Correlations of gut microbiota composition and serum levels of IL-1Β and IL-10 in the L-CRC, OB-CRC, and L-HC groups.

L-HC		L-CRC	OB-CRC	L-HC	L-CRC	OB-CRC
IL-1 B				IL-10		
*Blautia*	−0.621 (*p* = 0.024)	−0.812 (*p* = 0.021)	−0.656 (*p* = 0.024)	0.734 (*p* = 0.022)	0.912 (*p* = 0.003)	0.867 (*p* = 0.021)
*Roseburia*	−0.625 (*p* = 0.032)	−0.467 (*p* = 0.008)	−0.503 (*p* = 0.025)	0.865 (*p* = 0.008)	0.608 (*p* = 0.013)	0.854 (*p* = 0.017)
*Ruminoccocus*	−0.745 (*p* = 0.038)	−0.656 (*p* = 0.044)	−0.763 (*p* = 0.033)	0.898 (*p* = 0.005)	0.675 (*p* = 0.038)	0.854 (*p* = 0.018)
*Enterobacter*	0.843 (*p* = 0.015)	0.827 (*p* = 0.017)	0.834 (*p* = 0.015)	−0.892 (*p* = 0.011)	−0.912 (*p* = 0.021)	−0.895 (*p* = 0.012)
*Fusobacterium nucleatum*	0.865 (*p* = 0.187)	0.975 (*p* = 0.007)	0.965 (*p* = 0.009)	−0.878 (*p* = 0.523)	−0.997 (*p* = 0.003)	−0.898 (*p* = 0.003)
*Streptoccocus*	0.721 (*p* = 0.211)	0.815 (*p* = 0.0234)	0.834 (*p* = 0.0267)	−0.754 (*p* = 0.323)	−0.932 (*p* = 0.009)	−0.891 (*p* = 0.019)
*Enteroccocus faecalis*	0.674 (*p* = 0.252)	0.765 (*p* = 0.028)	0.793 (*p* = 0.017)	−0.763 (*p* = 0.237)	−0.911 (*p* = 0.015)	−0.870 (*p* = 0.029)
*Escherichia coli*	0.620 (*p* = 0.146)	0.645 (*p* = 0.019)	0.720 (*p* = 0.024)	−0.911 (*p* = 0.109)	−0.867 (*p* = 0.012)	−0.745 (*p* = 0.020)

**Table 3 ijms-21-06782-t003:** Correlations of gut microbiota composition and serum levels of zonulin and TMAO in the study groups.

	L-HC	L-CRC	OB-CRC		L-CRC	OB-CRC
Zonulin					TMAO	
*Ruminococcus*	−0.645 (*p* = 0.034)	−0.523 (*p* = 0.031)	−0.6490 (*p* = 0.031)	*Enterobacteraceae*	0.689 (*p* = 0.033)	0.632(*p* = 0.021)
*Prevotella*	0.445 (*p* = 0.443)	0.678 (*p* = 0.032)	0.858 (*p* = 0.033)	*Clostridium*	0.658 (*p* = 0.028)	0.778 (*p* = 0.020)
*Blautia*	−0.718 (*p* = 0.026)	−0.593 (*p* = 0.043)	−0.631 (*p* = 0.049)	*Streptococcus*	0.631 (*p* = 0.049)	0.593 (*p* = 0.043)
*Escherichia coli*	0.751 (*p* = 0.404)	0.545 (*p* = 0.019)	0.564 (*p* = 0.035)	*Escherichia coli*	0.763 (*p* = 0.021)	0.790 (*p* = 0.019)
*Desulfovibrio*	0.578 (*p* = 0.367)	0.748 (*p* = 0.035)	0.804 (*p* = 0.031)	*Desulfovibrio*	0.904 (*p* = 0.011)	0.7489 (*p* = 0.038)

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
