# Peer review of "Gut Microbiota-Mediated Inflammation and Gut Permeability in Patients with Obesity and Colorectal Cancer"

_ijms, 2020, doi:10.3390/ijms21186782_

Round 1

Reviewer 1 Report

Thank you for the opportunity to review "Gut microbiota-mediated inflammation and gut permeability in patients with obesity and colorectal cancer."

This is an important study using state-of-the-art methods and equipment. It brings it "all together" to address important questions about the gut microbiota.

The authors should address the fact that a tremendous number of statistical tests have been run which means that the significance levels of some of the comparisons and correlations may be due to chance (spurious). This is a problem with the area and not unique to this study. Given this the authors should also stress that the findings need to be replicated with larger samples and across different cultural/dietary groups.

Author Response

Comment:Thank you for the opportunity to review "Gut microbiota-mediated inflammation and gut permeability in patients with obesity and colorectal cancer."

This is an important study using state-of-the-art methods and equipment. It brings it "all together" to address important questions about the gut microbiota.

The authors should address the fact that a tremendous number of statistical tests have been run which means that the significance levels of some of the comparisons and correlations may be due to chance (spurious). This is a problem with the area and not unique to this study. Given this the authors should also stress that the findings need to be replicated with larger samples and across different cultural/dietary groups.

Response: We would like to thank the reviewer for their useful comments and suggestions that undoubtedly have helped to improve our manuscript. As suggested, in this new version of the manuscript we have added a paragraph relating to the reviewer's suggestions. Lines 474-476 in the revised manuscript.

Reviewer 2 Report

In the present study, the authors aimed to determine the intestinal microbiota composition in fecal samples from patients patients with colorectal cancer, with and without obesity compared to non-obese healthy controls, in order to unravel the possible association between the gut microbiota and microbial-derived metabolite TMAO, the inflammatory status, and the intestinal permeability in the context of obesity-associated CRC.

The authors have done very good work and the manuscript is overall well written. 

However, some points need to be addressed:

Lines 53-55: Sentence needs to be re-written

Line 465: Sentence needs to be re-written

Lines 474-476: Sentence needs to be re-written

Are there any available data from healthy individuals with obesity either from your collected data or from literature? That would add significant impact.

Author Response

Comment:In the present study, the authors aimed to determine the intestinal microbiota composition in fecal samples from patients with colorectal cancer, with and without obesity compared to non-obese healthy controls, in order to unravel the possible association between the gut microbiota and microbial-derived metabolite TMAO, the inflammatory status, and the intestinal permeability in the context of obesity-associated CRC.

The authors have done very good work and the manuscript is overall well written. 

Response:We really appreciate the general comment of reviewer 2

However, some points need to be addressed:

Comment:Lines 53-55: Sentence needs to be re-written

Response:The sentence in lines 53-55 has been re-written as suggested by the reviewer.

Comment:Line 465: Sentence needs to be re-written

Response:The sentence in line 465 has been re-written as indicated.

Comment:Lines 474-476: Sentence needs to be re-written

Response:The sentence in lines 474-476, now lines 476-479 has been re-written as suggested.

Comment:Are there any available data from healthy individuals with obesity either from your collected data or from literature? That would add significant impact.

Response: Thanks for this interesting idea. Unfortunately, at this time we do not have data on age- and gender-matched healthy obese patients (BMI ≥ 30 kg/m2) collected by our research group. After carefully checking in the literature, we have not found any study that includes a group of healthy obese individuals matched by race, sex and age with our recruited cohort of patients with either the CRC or the non-obese healthy control groups, and with a similar methodology to that used in our work (i.e. 16S rRNA sequencing and subsequent bioinformatic analysis, determination of serum levels of zonulin, TMAO and cytokines, etc). However, we agree that this additional group could add a significant impact to our studies so we will take it into account for future analyses.